# Hydroxyurea maintains working memory function in pediatric sickle cell disease

Jesyin Lai[1]*, Ping Zou[1], Josue L. Dalboni da Rocha[1], Andrew M. Heitzer[2],
Tushar Patni[3], Yimei Li[3], Matthew A. Scoggins[2], Akshay Sharma[4], Winfred C. Wang[5],
Kathleen J. Helton[1], Ranganatha Sitaram[1]*

1 Department of Diagnostic Imaging, St. Jude Children's Research Hospital, Memphis, Tennessee, United States of America, 2 Department of Psychology and Biobehavioral Sciences, St. Jude Children's Research Hospital, Memphis, Tennessee, United States of America, 3 Department of Biostatistics, St. Jude Children's Research Hospital, Memphis, Tennessee, United States of America, 4 Department of Bone Marrow Transplantation & Cellular Therapy, St. Jude Children's Research Hospital, Memphis, Tennessee, United States of America, 5 Department of Hematology, St. Jude Children's Research Hospital, Memphis, Tennessee, United States of America

⊕ These authors contributed equally to this work.
* jesyin.lai@stjude.org (JL); ranganatha.sitaram@stjude.org (RS)

**Data Availability Statement:** Public deposition of data would breach compliance with the protocol approved by St. Jude Children's Research Hospital IRB. Therefore, public data availability would compromise patient privacy. The raw data, with

## Abstract

Sickle cell disease (SCD) decreases the oxygen-carrying capacity of red blood cells. Children with SCD have reduced/restricted cerebral blood flow, resulting in neurocognitive deficits. Hydroxyurea is the standard treatment for SCD; however, whether hydroxyurea influences such effects is unclear. A key area of SCD-associated neurocognitive impairment is working memory, which is implicated in other cognitive and academic skills. The neural correlates of working memory can be tested using n-back tasks. We analyzed functional magnetic resonance imaging (fMRI) data of patients with SCD (20 hydroxyurea-treated patients and 11 controls, aged 7–18 years) while they performed n-back tasks. Blood-oxygenation level–dependent (BOLD) signals were assessed during working memory processing at 2 time points: before hydroxyurea treatment and ~1 year after treatment was initiated. Neurocognitive measures were also assessed at both time points. Our results suggested that working memory was stable in the treated group. We observed a treatment-by-time interaction in the right cuneus and angular gyrus for the 2- >0-back contrast. Searchlight-pattern classification of the 2 time points of the 2-back tasks identified greater changes in the pattern and magnitude of BOLD signals, especially in the posterior regions of the brain, in the control group than in the treated group. In the control group at 1-year follow-up, 2-back BOLD signals increased across time points in several clusters (e.g., right inferior temporal lobe, right angular gyrus). We hypothesize that these changes resulted from increased cognitive effort during working memory processing in the absence of hydroxyurea. In the treated group, 0- to 2-back BOLD signals in the right angular gyrus and left cuneus increased continuously with increasing working memory load, potentially related to a broader dynamic range in response to task difficulty and cognitive effort. These findings suggest that hydroxyurea treatment helps maintain working memory function in SCD.

participant identifiers removed, supporting the conclusions of this study will be accessible to qualified researchers upon request, by contacting the St. Jude's Human Research Protection Program at Elroy.Fernandes@stjude.org.

**Funding:** This research was supported, in part, by the American Lebanese Syrian Associated Charities (ALSAC) in the form of salaries to all of the authors, equipment, and other expenses. The National Heart, Lung, and Blood Institute provided support in the form of grants to AMH [K23HL166697] and AS [1U01HL163983]. The content is solely the responsibility of the authors and does not necessarily represent the official views of the National Institutes of Health.

**Competing interests:** Dr. Akshay Sharma has received consultant fees from Spotlight Therapeutics, Medexus Inc., Vertex Pharmaceuticals, Sangamo Therapeutics, and Editas Medicine. He has also received research funding from CRISPR Therapeutics and honoraria from Vindico Medical Education. Dr. Sharma is the St. Jude site principal investigator of clinical trials for genome editing for SCD sponsored by Vertex Pharmaceuticals/CRISPR Therapeutics (NCT03745287), Novartis Pharmaceuticals (NCT04443907), and Beam Therapeutics (NCT05456880). The industry sponsors provide funding for the clinical trial, which includes salary support paid to our institution. Dr. Sharma has no direct financial interest in these therapies. Dr. Andrew M. Heitzer has received consulting fees from Global Blood Therapeutics. Neither Dr. Sharma nor Dr. Heitzer is an employee of any commercial entity. All other authors declare that the research was conducted in the absence of any commercial or financial relationships that could be construed as a potential conflict of interest. This does not alter our adherence to PLoS ONE policies on sharing data and materials.

# 1. Introduction

Sickle cell disease (SCD) is a devastating hematologic disease marked by acute and chronic cerebrovascular changes. Patients with SCD often have symptoms of brain injury, including neurocognitive deficits [1, 2]. Without intervention, approximately 11% of patients who have the most severe form of SCD [hemoglobin (Hb) SS] experience overt stroke before 20 years of age [3]. Moreover, 17%-22% of patients with SCD experience silent cerebral infarction (SCI), defined as focal ischemic damage on conventional magnetic resonance imaging (MRI) without clinical signs or symptoms of stroke [1, 4, 5]. SCI is an independent risk factor for overt stroke and is associated with lower scores on math and reading achievement, Full-Scale Intelligence Quotient (FSIQ), Verbal IQ, and Performance IQ tests [4–6].

In patients with SCD, neurocognitive deficits arise during early childhood and persist into adulthood, negatively affecting their health outcomes and quality of life and increasing disease complications [7]. Neurocognitive deficits are also exacerbated by environmental factors, such as low socioeconomic status [8]. In a meta-analysis of 110 studies involving 3600 participants with SCD, deficits in FSIQ, verbal reasoning, perceptual reasoning, and executive function increased from preschool- to school-aged participants [9]. Furthermore, more severe anemia, lower total Hb and fetal Hb, cerebral occlusion, and reduced oxygen saturation are all associated with more severely impaired cognitive processing in SCD [10]. In some cases, however, neurocognitive deficits occur without neurologic injury, i.e., in individuals with normal diagnostic imaging [9, 11].

The current standard treatment for SCD is hydroxyurea. Patients receive daily oral administration of hydroxyurea, often throughout their lifetimes. Several studies have attempted to identify the effects of hydroxyurea treatment on neurocognitive performance in children with SCD [12–15]. Among these, Puffer et al. (2007) reported an improved global cognitive index with hydroxyurea treatment, and Heitzer et al. (2021) found an association of hydroxyurea therapy with higher neurocognitive measures across multiple assessments. Wang et al. (2021) showed significantly improved reading comprehension after hydroxyurea treatment in patients with SCD. Fields et al. (2019) obtained neuroimages of patients with SCD and found lower cerebral oxygen–extraction fraction in patients receiving hydroxyurea than in those who were not [16]. Nottage et al., (2016) reported children with SCD receiving hydroxyurea for 3–6 years have a low rate of new or worsening cerebrovascular disease [16, 17].

It is important that we understand the neural correlates of SCD-associated deficits, the treatment effects of hydroxyurea on brain function, and the associated mechanisms. Task-based functional magnetic resonance imaging (fMRI) approaches have been developed to investigate the brain regions involved in SCD-related neurocognitive deficits. Among them, working memory is a critical area of impairment, because it substantially affects quality-of-life outcomes, including academic development and achievement [18, 19]. Furthermore, working memory is commonly affected in youths with SCD [20]. Brain regions with working memory functions are relatively more vulnerable to SCD than are other regions. For instance, oxygen delivery to deep white matter, basal ganglia, middle and superior frontal gyrus, and dorsal parietal regions is often disrupted in patients with SCD due to derangement in cortical and subcortical structures supplied by the distal portions of the anterior and middle cerebral artery [2, 21]. To assess working memory, cognitive neuroscientists use the n-back test [22]. During an n-back test, participants are instructed to monitor a series of stimuli and respond by button pressing to a stimulus presented n trials earlier [23, 24]. By using a computerized visual–spatial n-back task, some studies have shown evidence of poorer visual–spatial working memory in children with SCD, compared to demographically matched healthy controls [20, 25]. To identify the brain regions involved in working memory processing and its deficits in patients with

SCD and to trace the longitudinal neurocognitive effects of hydroxyurea treatment, we must investigate task-based fMRI at baseline and follow-up time points.

In this study, we hypothesized that hydroxyurea improves working memory and influences working memory processing in the brains of patients with SCD. We also aimed to identify hydroxyurea treatment effects in brain regions related to working memory processing. To test our hypothesis, we acquired fMRI data during n-back tasks using images of objects as stimuli in pediatric patients with SCD before initiating hydroxyurea treatment and at 1-year follow-up. We also analyzed brain-activation patterns during working memory processing and identified regions with changes in activation at 1-year follow-up via searchlight-pattern classification. Our results support a potential role of hydroxyurea in maintaining/stabilizing working memory processing and preventing it from worsening in patients with SCD.

## 2. Methods

### 2.1 Participants

Patients (7–18 years old) who had a diagnosis of SCD (HbSS or $HbS\beta^0$-thalassemia), had not been previously exposed to hydroxyurea, and met clinical criteria to receive this therapy were recruited to participate in this study. Patients with other forms of SCD (HbSC, $HbS\beta^+$-thalassemia) usually have milder disease and less severe symptoms [26] and their neurocognitive function may differ from that of those with HbSS. To have a more homogenous population, patients with HbSC or $HbS\beta^+$-thalassemia were excluded. In addition, participants were excluded if they had a history of clinical stroke, were receiving chronic transfusion therapy, had received a hematopoietic stem cell transplant, or could not tolerate MRI examination without sedation or anesthesia. Participants were recruited from the Sickle Cell Outpatient Clinic at St. Jude Children's Research Hospital (St. Jude) between June 2011 and January 2015, when they were initiating hydroxyurea therapy.

This study was approved by the St. Jude Institutional Review Board. Written informed consent was obtained from each participant or from the legal guardian of each minor participant. Each minor participant also gave verbal assent before participating in the research examinations.

### 2.2 Study design

Participants were evaluated before starting hydroxyurea therapy. Baseline MR imaging, transcranial Doppler ultrasound examination, neurocognitive testing, and blood work were completed, usually over 2 days. In the treated group, hydroxyurea dosing was initiated at 20 mg/kg/day and gradually escalated per the standard protocol [27]. To ensure hydroxyurea was used as recommended, we monitored prescription refills and assessed laboratory indicators of hydroxyurea compliance, including mean corpuscular volume, reticulocyte count, etc. All evaluations were conducted at 2 time points, baseline before hydroxyurea treatment commenced and at 1-year follow-up. Participants were classified as those who received hydroxyurea treatment and those who did not.

A total of 31 patients with SCD were approached, consented, and enrolled in the study. Although all participants completed subsequent experimental tasks while simultaneously undergoing scanning at both time points, some participants' data at one or both time points were excluded due to imaging quality concerns. For the HU group, data from 4 participants were excluded at the pre-treatment time point and data from 3 were excluded at the 1-year follow-up. In the non-HU control group, data from 3 participants were excluded at the 1-year follow-up.

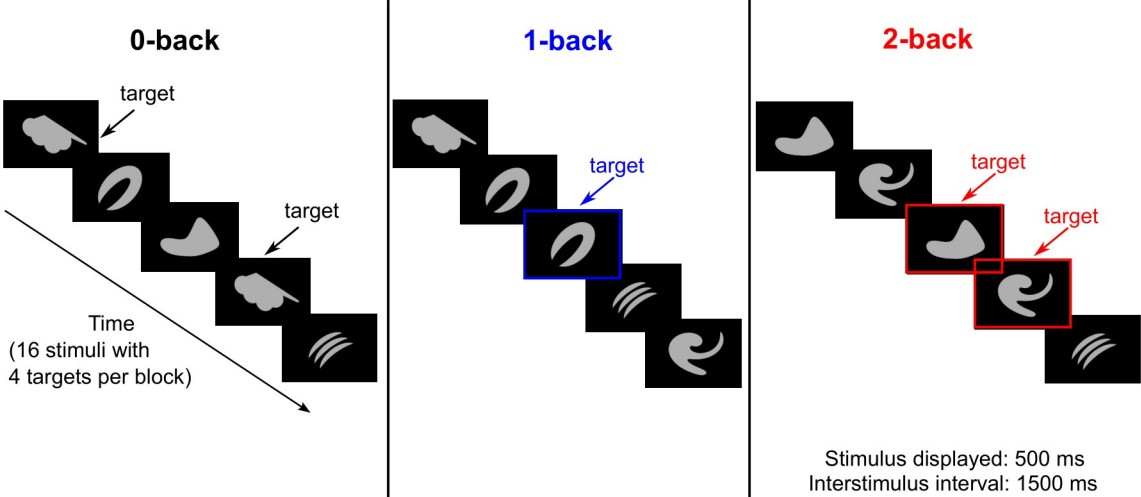

**Fig 1. Schematic overview of the n-back task using images of objects as stimuli.** The 0-, 1-, and 2-back tasks were performed separately in each block.

## 2.3 Functional magnetic resonance imaging

Siemens MRI 3T scanners were used in this study. A 3-dimensional (3D) T1-weighted sagittal MPRAGE sequence (TR/TE 1980 ms/ 2.26 ms, 1.0-mm pixel space, 256 × 256 matrix, 160 slices, and 1.0-mm slice thickness) was acquired to facilitate spatial processing of functional images and visualize fMRI results. The fMRI examinations were performed with a T2*-weighted echo-planar imaging sequence with the following parameters: TR = 2.0 s, TE = 30 ms, flip angle = 90˚, field of view = 192 mm × 192 mm, matrix = 64 × 64, slice thickness = 3.5 mm, and a total of 32 slices in 1 volume. Participants were trained for the fMRI-scanning sessions with computerized programs before entering the scanner room. The patient's head was immobilized with vacuum pads in the head coil during the scanning sessions, and fMRI instructions and stimuli were viewed through the head coil mirror.

## 2.4 Object n-back tasks

Participants were tested with 0-, 1-, and 2-back tasks using objects (i.e., abstract images not easily namable, Fig 1) as stimuli, while undergoing fMRI scanning. Each n-back task was repeated 3 times with the order of 0-, 1-, and 2-back (i.e., total = 9 blocks). For the 0-back task, participants responded by pressing a button when they detected a specified target (Fig 1). For the 1- and 2-back tasks, participants responded if the current image was identical to the previous image (1-back) or the image presented 2 trials earlier (2-back). Before each task, instructions associated with each n-back task were displayed for 2-s, followed by a 5-s rest period. During each task block, a series of 16 stimuli, including 4 targets, was presented. Each stimulus was displayed for 500 ms, with a 1500-ms interval between stimuli. Reaction time, omissions, and commissions were recorded during all n-back tasks. In addition, balanced accuracy (BA), which is the mathematical mean of sensitivity and specificity, was measured using Eq 1 (Eq 1) below. BA was used instead of "overall accuracy" because the total numbers of targets (n = 12)

and nontargets (n = 36) were imbalanced.

$$BA = \frac{1}{2} \times \left[ \left( \frac{hits}{total\ targets} \right) + \left( \frac{correct\ rejections}{total\ nontargets} \right) \right] \tag{1}$$

### 2.5 Neurocognitive tests

Neurocognitive measures sensitive to the deficits in SCD were chosen and tested [9]. Measures were chosen with appropriate test-retest reliability and negligible practice effects for the 1-year interval between testing time points. The intellectual function of participants was tested with the Wechsler Intelligence Scale for Children, Fourth Edition (WISC-IV) [28]. The WISC was used to assess intelligence in children 7 to <18 years of age, and the Wechsler Adult Intelligence Scale, Fourth Edition (WAIS-IV) [29] was used for participants 18 years and older. A shortened administration of 8 subtests enabled the derivation of the index scores of FSIQ, verbal comprehension, perceptual reasoning, working memory, and processing speed.

### 2.6 fMRI pre-processing and analysis

The fMRI data were pre-processed and analyzed with SPM software (www.fil.ion.ucl.ac.uk/spm). For first-level analyses, SPM 8 was used; for group-level analyses, SPM 12 was used. Motion-corrected fMRI time series from the MRI scanner were used in fMRI analysis. Pre-processing included slice-timing correction (i.e., corrects differences in image acquisition time between slices) [30], realignment (i.e., realigns a time series of images acquired from the same subject), co-registration (i.e., performs within-subject registration using a rigid-body model) to the 3D-T1 image [31], spatial normalization (i.e., matches an individual scan to a template) to the Montreal Neurological Institute (MNI) brain template, and spatial smoothing (i.e., suppresses noise and effects due to residual differences during inter-subject averaging) [32] with an 8-mm$^3$ FWHM (full-width, half-maximum) Gaussian kernel. Individual contrast maps from the first-level analysis were used in a full-factorial second-level analysis model to assess the group effects, time point effects, and group–time interaction effects.

### 2.7 Searchlight analysis

The details of searchlight analysis were presented by Kriegeskorte et al. (2006) [33]. In this method, the imaged volume of the brain is scanned with a "searchlight," and the contents undergo multivariate classification at each location in the brain. In brief, a 3D sphere with a given radius (usually in millimeters) is scanned across the whole brain or regions of interest. A classifier or regressor is then trained on the corresponding voxels to predict the experimental, relevant behavior or condition, and prediction accuracy is measured for each corresponding voxel.

We performed searchlight analysis with a software program developed in-house in the Python language using the "nilearn" package [34]. During the analysis, we first isolated object-evoked fMRI signals during the 2-back task measured at baseline and at 1-year follow-up from participants with fMRI data at both time points. Because we performed searchlight analysis on fMRI data to classify time points, those participants with fMRI data at only 1 time point were excluded to avoid data imbalance. Among the enrolled patients, 13 participants in the treated group and 8 participants in the control group had fMRI data at both time points. At each time point, BOLD signals of 2-back tasks were obtained by subtracting the average 0-back object-evoked fMRI signals from each trial of 2-back object-evoked fMRI signals. We retrieved 42

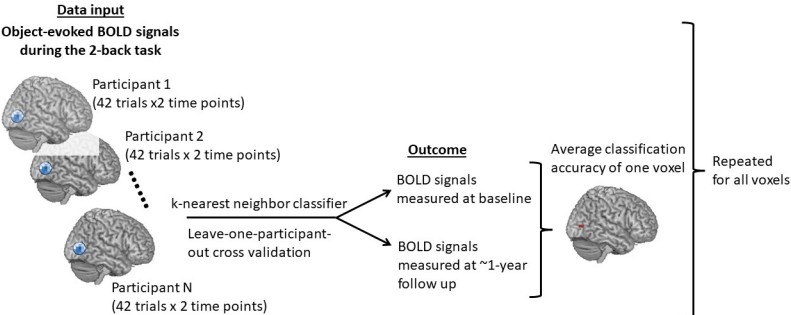

**Fig 2. Schematic overview of the searchlight-pattern classification.** Similar classifications were conducted in the HU and non-HU groups separately.

trials of BOLD signals in the 2-back task at each time point. Next, we performed searchlight analysis over the whole brain to identify the areas containing information to classify BOLD signals into those measured at pre-treatment vs. 1-year follow-up. The same searchlight analysis was conducted separately in the treated and the control groups. We used a sphere radius of 4 mm, leave-one-participant-out cross validation, and a k-nearest neighbor classifier [35]. According to Kriegeskorte et al. (2006) [33] and Etzel et al. (2013) [36], a 4-mm radius usually yields near-optimal detection performance. After testing different classifiers, we concluded that the k-nearest neighbor classifier provided the best classification accuracy. Neural discrimination accuracy using this classifier was also reported to be highly correlated with animal behavior [37, 38].

Fig 2 illustrates a complete process of searchlight analysis within the treated or the control group. The searchlight analysis produced an accuracy score (0–1.0) for each voxel: 0 indicates no predictive power and 1.0 indicates best predictive power. To identify specific brain regions that contained information with higher accuracies to distinguish BOLD signals measured at pre-treatment vs. 1-year follow-up, we applied the Python package "AtlasReader" [39] to identify significant clusters and the anatomical locations of these clusters. A cut-off threshold for an accuracy score of 0.65, a threshold for a cluster size of 20 voxels, and the AAL (Automated Anatomical Labeling) Atlas [40] were used. Among all the clusters derived from searchlight analyses in each group, we selected clusters with volumes $\geq$520 mm$^3$ and peak accuracies $\geq$0.7 for subsequent analyses. The median of BOLD signals in each selected cluster at each time point and during different levels of n-back tasks were computed for comparisons. The BOLD signal differences between time points in each voxel of each selected cluster were also computed for subsequent analyses of brain/behavior relations.

## 2.8 Statistical analyses

We compared n-back task performance (including reaction time, BA, commission errors, and omission errors), neurocognitive measures, and the medians of 2-back BOLD signals for each selected cluster across time points (within group) or across groups. For n-back task performance and the medians of 2-back BOLD signals, the Wilcoxon rank-sum test was used to compare variables between the 2 groups at a single time point. The Wilcoxon signed-rank test was also used to compare variables between the 2 time points within the same group. For neurocognitive measures, the Student's *t*-test was used for between-group comparison at a single time point, and the paired-sample *t*-test was used for within-group comparison across time points. In addition, we compared the median BOLD signals for each selected cluster at the

1-year follow-up time point as the working memory load increased from 0- to 2-back. The Wilcoxon signed-rank test was used to compare BOLD signals between task difficulties within the same group. All these analyses were exploratory in nature due to the limited number of observations in each condition. As a result, we did not adjust analyses for multiple comparisons. For these analyses, *p*-values were 2-sided, and $p < .05$ was considered statistically significant.

## 2.9 Brain and behavior relations

We explored the association between 2-back BOLD signal differences (i.e., the 1-year follow-up data were subtracted from the pre-treatment data) and behavior performance differences of the 2-back task for each selected cluster. A significant increase in omission errors in the 2-back task was found in the non-HU control group across time points (Fig 3D), we focused on

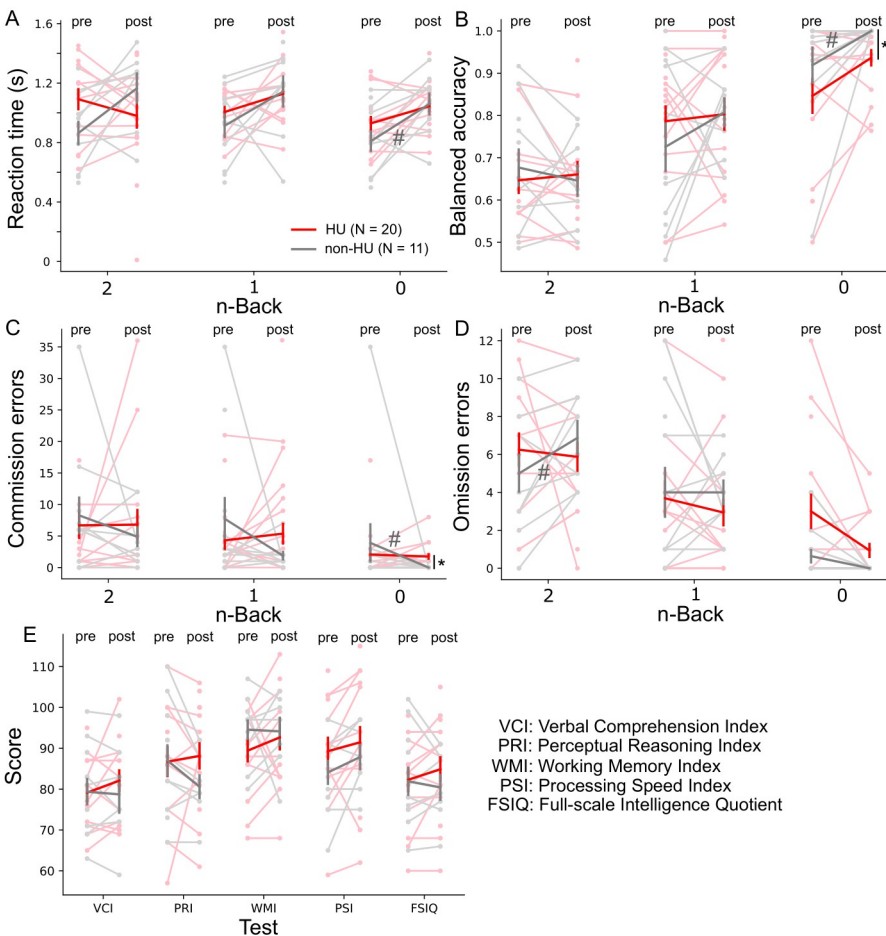

**Fig 3. No change in performance on the 0- to 2-back tasks or on the neurocognitive tests with hydroxyurea treatment.** Reaction time (**A**), balanced accuracy (**B**), commission errors (**C**), and omission errors (**D**) of the 0-, 1-, and 2-back tasks were measured pre-treatment and at 1-year follow-up. Neurocognitive tests (**E**) were also measured at both time points. When compared across the hydroxyurea (HU) and non-HU control groups at the 1-year follow-up time point, the balanced accuracy and commission errors of the 0-back task significantly differed. Red and gray bold lines represent the means of the HU and the non-HU control groups, respectively. Light red and light gray markers represent individual data points in the HU and the non-HU control groups, respectively. * $p < .05$ (Wilcoxon rank-sum test) and # $p < .05$ (Wilcoxon signed-rank test).

investigating the omission error change and BOLD signal change between the 2 time points for the 2-back task. Only data from participants with data on omissions errors and BOLD signals at both time points were used in this analysis. We fitted a regression line (Eq 2) with BOLD signal differences (*BOLD_diff*), omission error differences (*OM_diff*), and group (*grp*: non-HU control = 0 and HU = 1) for each of the selected clusters separately.

$$E(BOLD\_diff \,|\, OM\_diff,\, grp) = \beta_0 + \beta_1 * OM\_diff + \beta_2 * grp + \beta_3 * OM\_diff * grp \qquad (2)$$

In the above equation, $\beta_0$ is the intercept, $\beta_1$, and $\beta_2$ are the coefficients for differences in behavior performance between time points and group, respectively, and $\beta_3$ is the coefficient for the interaction of behavior differences and group. We focused on the significance of the slopes (i.e., *β1* for the non-HU control group and *β1+β3* for the HU group) of the fitted regression lines for each selected cluster in each group. The slopes (*β1* and *β1+β3*) indicate the change in *BOLD_diff* with a unit change in *OM_diff* in the non-HU control group and the HU group, respectively.

## 3. Results

### 3.1 Participants and hydroxyurea treatment

Eleven participants were assigned to the non-HU control group because they either never initiated hydroxyurea treatment or discontinued the drug shortly after starting it. The mean age of the controls was 11.9 years (SD 2.5 years, range 7.6–16.5 years) at enrollment; 3 (27%) were female. The 20 patients who received hydroxyurea therapy continuously were assigned to the HU group. These participants had a mean age of 12.3 years (SD 3.3 years, range 7.1–17.3 years) at enrollment; 12 (60%) were female.

For the HU group, the initial hydroxyurea dose was 20 mg/kg/day, and doses were gradually adjusted to an average of 23.8 mg/kg/day (range 13.3–33.4 mg/kg/day) after 1 year of treatment. Hematologic responses to hydroxyurea at 1 year suggested good adherence to treatment [15]. The Hb level, fetal Hb level, absolute reticulocyte count, and other measures have been previously reported [15] and can be found in S1 Table. The comparison of baseline 1-year follow-up measurements revealed a notable response to hydroxyurea treatment, despite relatively constrained duration of dosing at the maximum-tolerated dose. During the 1-year period, no cerebral vascular accidents were reported among the participants. However, 47% of participants exhibited SCIs at their initial examination, with the incidence rising to 53% upon conclusion of the study (refer to section 3.3.1 of [15] for details).

### 3.2 Performance of n-back tasks and neurocognitive measures

We measured the longitudinal effects of hydroxyurea treatment on performance of the n-back tasks and neurocognitive tests. We did not observe a significant treatment effect within the HU group (Fig 3A–3D; Wilcoxon signed-ranked test, $p > .05$) when all n-back tasks were compared across time points. Within the non-HU control group, we found significant differences in the reaction time, BA, and commission errors for the 0-back task, when compared across time points. The omission errors increased significantly for the 2-back task in the control group, when compared across time points. At the 1-year follow-up, we did not observe differences between the HU and non-HU control groups (Wilcoxon rank-sum test, $p > .05$), except in the BA and commission errors of the 0-back task. Although there were more participants in the HU group with the neurocognitive measures, we did not detect a significant difference

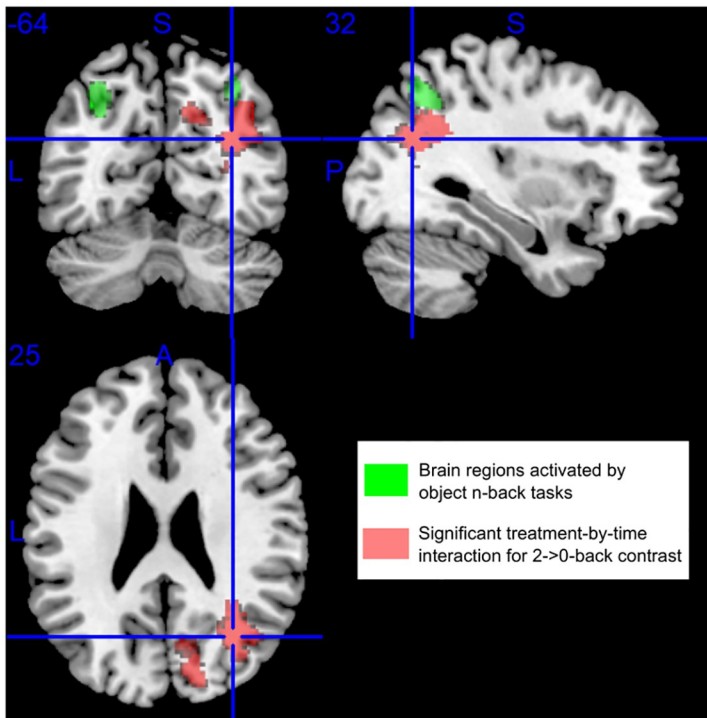

**Fig 4. A significant treatment-by-time interaction was detected in the right cuneus and angular gyrus (labeled in red) for 2- >0-back contrast.** The brain regions activated by the n-back tasks using objects as stimuli are labeled in green. Blue lines mark the Montreal Neurological Institute (MNI) coordinate [32, −64, 25].

between the 2 groups (Student's *t*-test, *p* >.05) or the 2 time points (paired-sample *t*-test, *p* >.05) (Fig 3E).

### 3.3 fMRI interaction effect of the treatment group and time point

We tested various contrasts, including 1- >0-back, 2- >1-back, 2- >0-back, and 2-back > resting-condition by using full factorial modeling with both groups and time points to search for main and interaction effects. We also compared, for example, 2- >0-back contrast between pre-treatment and ~1-year follow-up in the HU and non-HU control groups separately. Among all the tests and comparisons, we detected a significant interaction effect ($T_{48}$ = 3.27, *p* = .0001) of the treatment group and time point in the right cuneus and angular gyrus only for the 2- >0-back contrast in participants from both groups (Fig 4). The main and interaction effects of the treatment group and time point were not significant when we used the other contrasts.

### 3.4 Brain regions of searchlight-pattern classification

We did not observe significant treatment effects with hydroxyurea when performing traditional fMRI univariate analysis; thus, we shifted our analytical strategy to multivariate pattern analysis. Searchlight-pattern classification is a type of multivariate classification at each location in the brain that helps identify regions predictive of experimental conditions [33]. We performed searchlight analyses to identify and localize brain regions or clusters that predicted 2-back BOLD signals recorded at pre-treatment or 1-year follow-up. These analyses were

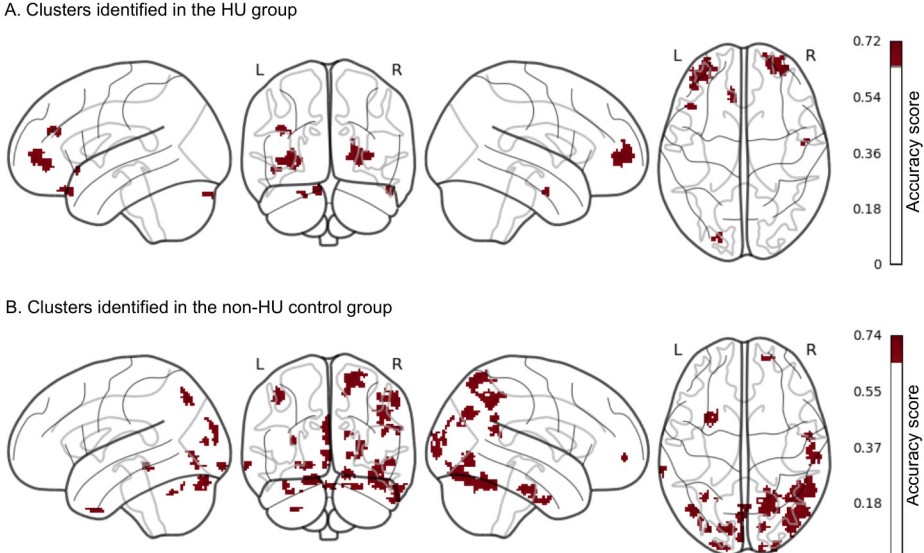

**Fig 5. More clusters with changes in brain activation between time points during the 2-back task were identified in the non-HU control group than in the HU group.** Clusters with higher predictive power of time points were identified using a classification accuracy threshold of >.65 and a cluster-size threshold of >20 voxels. These clusters had larger differences in 2-back BOLD signals between time points. In the non-HU group, we identified 27 clusters, with most larger clusters localizing in the posterior region In the HU group, we identified 7 clusters with only 2 larger clusters localizing in the frontal region.

conducted separately in the HU group and the non-HU group. Brain regions or clusters with higher classification accuracy scores were those with more prominent changes in brain activation between time points during the 2-back task. Clusters with classification accuracies >.65 and size >20 voxels identified in each group are shown in Fig 5. A total of 27 clusters, with most larger clusters localizing in the posterior region, were reported in the non-HU group. In contrast, 7 clusters, with only 2 larger clusters localizing in the frontal region were reported in the HU group. Next, we selected clusters with peak accuracy $\geq$.7 and volume $\geq$520 mm$^3$ for downstream analyses. Among all the identified clusters in both groups, 7 in the non-HU group and 2 in the HU group fulfilled the selection criteria. The anatomical locations and the main brain regions involved in these 9 selected clusters are shown in Fig 6. Moreover, the details of these clusters, including peak MNI coordinates, volume, brain regions with percentage of coverage in the AAL Atlas, etc., are reported in S2 (control group) and S3 (HU group) Tables.

## 3.5 Brain activation during working memory processing

For each of the 9 selected clusters, we computed the median of 2-back BOLD signals at the 2 time points, respectively, in both groups (Fig 7A). In the non-HU control group, 6 of the selected clusters (corresponding to the right crus I cerebellum, right inferior parietal lobe, right inferior temporal lobe, right angular gyrus, left cuneus, and left middle frontal gyrus) had higher BOLD signals during working memory processing at 1-year follow-up ($p < .05$, Wilcoxon signed-rank test). However, only 1 of the selected clusters (the right superior parietal lobe) had higher BOLD signals in the HU group at 1-year follow-up ($p < .05$, Wilcoxon signed-rank test). We also computed the median of BOLD signals for each selected cluster during 0-, 1-, or 2-back tasks at 1-year follow-up (Fig 7B). In the non-HU control group, BOLD

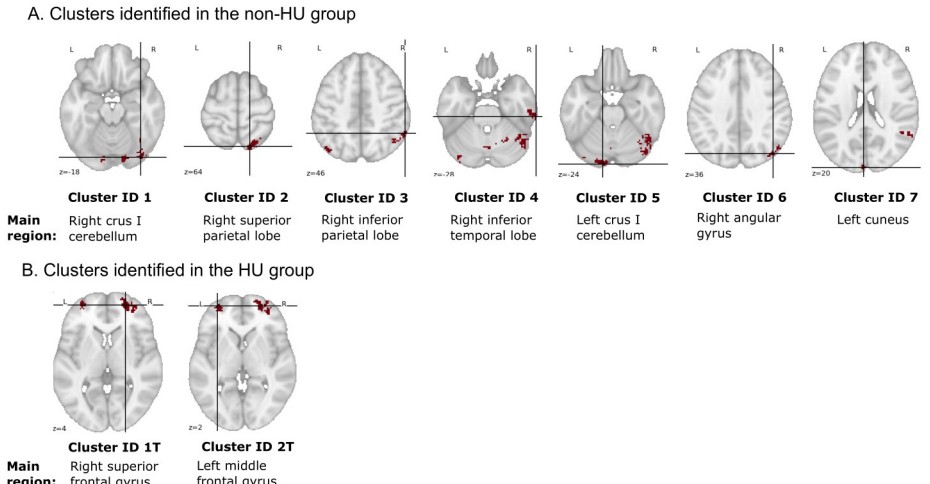

**Fig 6. Anatomical locations of the 9 selected clusters in the transverse plane.** Clusters with peak accuracy $\geq 0.7$ and volume $\geq 520$ mm$^3$ were selected for downstream analysis. The respective main region in the brain (the highest percentage coverage in the AAL Atlas) for each cluster is shown underneath. The details of these clusters are reported in S2 and S3 Tables. The crossed lines mark the peaks of the clusters.

signals elevated as the working memory load increased from 0- to 1-back ($p < .05$, Wilcoxon signed-rank test), but not from 1- to 2-back, in the right inferior temporal lobe, right angular gyrus, and right superior frontal gyrus. However, BOLD signals continued to increase ($p < .05$, Wilcoxon signed-rank test) in the left crus I cerebellum, right angular gyrus, left cuneus, and right superior frontal gyrus of the HU group as the working memory load increased from 0- to 2-back. In summary, 2-back BOLD signals increased across time points, and with increasing working memory load from 0- to 1-back, BOLD signals increased in the non-HU control group at 1-year follow up. Meanwhile, BOLD signals increased continuously with increasing working memory load, from 0- to 2-back in the HU group at 1-year follow up.

We analyzed the relations of differences in 2-back BOLD signals and differences in omission errors between time points for each selected cluster. These analyses were exploratory in nature, as we had a limited number of participants ($n = 8$ in the non-HU control group and $n = 12$ in the HU group) with fMRI data and omission errors at both time points. Among all the fitted regression lines (*BOLD_diff* vs. *OM_diff*), we did not observe significant differences in the slopes in either group, which indicates we did not find that the difference of the BOLD signal between 2 time points was associated with the omission error changes between the same time points in both groups.

## 4. Discussion

In this study, we infer stability in working memory processing in patients who received hydroxyurea treatment and worsening of that measure in those in the control group since we did not observe significant differences in behavior measures of all n-back tasks in the HU group. In the fMRI group-level analysis of 2- > 0-back contrast, we detected increased BOLD signals in the HU group but reduced BOLD signals in the non-HU control group in the right cuneus and angular gyrus when compared between baseline and 1-year follow-up time points. In addition, via searchlight-pattern analyses, we identified more changes in the pattern and magnitude of BOLD signals in the posterior part of the brain across time points in the non-HU control group than in the HU group. In the control group, 2-back BOLD signals increased

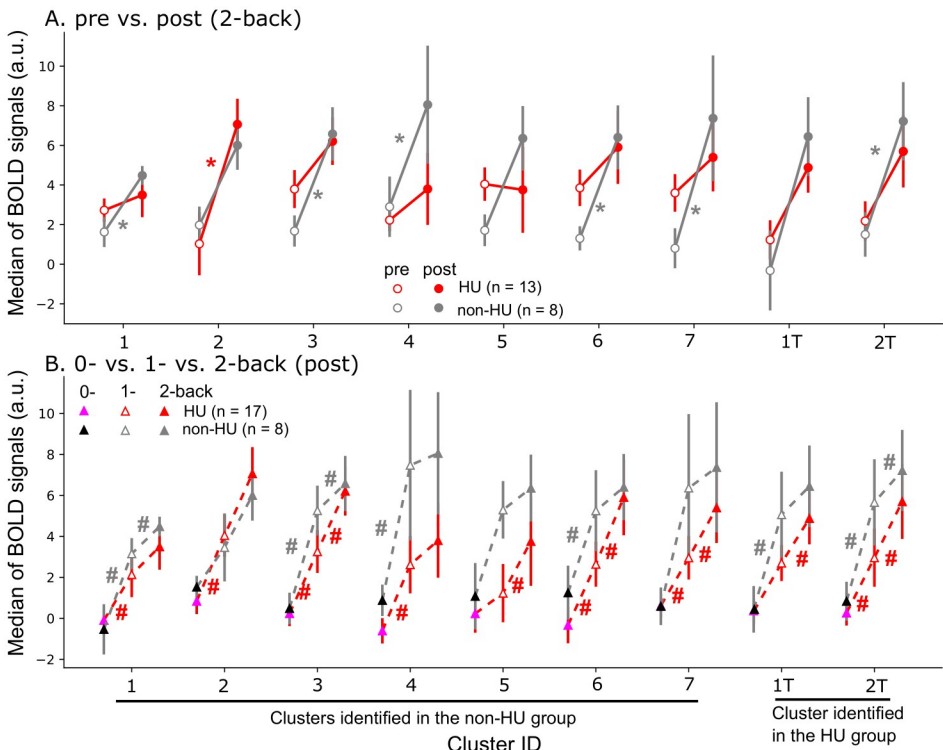

**Fig 7. Increased BOLD signals during working memory processing as a function of time points and task difficulty.** Nine clusters were selected from all the clusters identified via searchlight analyses. (A) Six selected clusters in the non-HU control group had higher brain activation during working memory processing at 1-year follow-up than at pre-treatment. However, only 1 cluster had higher brain activation in the HU group at 1-year follow-up compared to pre-treatment. * $p < .05$ [Wilcoxon signed-rank test within the HU group (red) or the non-HU control group (gray)]. (B) Increased BOLD signals were observed in most selected clusters in both groups, as the working memory load increased from 0- to 1-back. In the non-HU control group, BOLD signals did not increase further when the working memory load increased from 1- to 2-back. In the HU group, however, BOLD signals increased continuously as the working memory load increased from 0- to 2-back. # $p < .05$ (Wilcoxon signed-rank test).

across time points and BOLD signals elevated with increasing working memory load from 0- to 1-back in clusters, e.g., the right inferior temporal lobe, right angular gyrus, etc., at 1-year follow-up, which may result from increased cognitive effort during working memory processing with no hydroxyurea treatment [41]. In the HU group, BOLD signals in the right angular gyrus, left cuneus, etc. increased continuously with increasing working memory load from 0- to 2-back, potentially related to a broader dynamic range in response to task difficulty and cognitive effort [41]. Together, these findings suggest that hydroxyurea treatment helps maintain working memory function in SCD.

## 4.1 Effect of hydroxyurea treatment on cognition

Hydroxyurea treatment is a standard of care for children with SCD, and several studies have shown positive effects of hydroxyurea treatment on neurocognitive functions. One study reported improved verbal comprehension, global cognitive ability, and fluid reasoning in 15 patients with SCD (aged 6–21 years) after at least 1 year of hydroxyurea treatment compared to 50 untreated patients [12]. Our recent study showed a notable trend of improved FSIQ and a significant gain in reading comprehension in 19 patients with SCD (aged 7.2–17.8

years), comparing before and after 1 year of hydroxyurea treatment [15]. Moreover, another cross-sectional study assessing neurocognitive performance in 364 patients with SCD of multiple age groups (8–24 years) revealed that early treatment with hydroxyurea potentially reduces, but does not eliminate, neurocognitive decline with age [13].

The mechanism of how hydroxyurea improves cognitive outcomes is not completely clear [12]. One potential mechanism is that hydroxyurea suppresses cell-stress signaling, leading to increased levels of fetal hemoglobin and hemoglobin [42], which cause positive effects on brain oxygenation and function. As hydroxyurea decreases anemia and thrombocytosis [43, 44], it may prevent or reverse functional brain abnormalities, slowing the process of neurocognitive declines. In addition to affecting hemoglobin level and brain oxygenation, hydroxyurea may lead to improved cognitive performance more indirectly by decreasing general illness severity and pain [45].

In this study, hydroxyurea treatment did not affect behavioral measures of reaction time, BA, omission errors, or commission errors, nor did it affect any of the neurocognitive measures as assessed by fMRI. However, in patients receiving no hydroxyurea treatment, increased omission errors in the 2-back task were observed when compared across time points within the group. Moreover, changes in reaction time, BA and commission errors were found within the non-HU control group when compared across time points. These observations may be due to stability in working memory processing in patients with hydroxyurea treatment but more fluctuations in those with no hydroxyurea treatment. The lack of significant differences with hydroxyurea treatment in other neurocognitive measures and the n-back tasks could also be because a longer time interval (i.e., more than 1 year) between assessments is required to detect significant changes in behavior performance of working memory and neurocognitive tests. Furthermore, the neurocognitive effects of hydroxyurea may depend on when patients start the treatment. Hydroxyurea may produce greater neurocognitive effects in patients who start the treatment earlier in life, during the critical period of development, or in those with a lower baseline ability [15].

Although we observed no changes in most behavior measures of the 2-back task, we identified changes in brain activation during the 2-back task via searchlight-pattern analyses in both group. Searchlight-pattern classification of 2-back BOLD signals into those measured at baseline vs. 1-year follow-up reported more clusters with higher accuracy scores ($>.65$) in the non-HU control group than in the HU group. We also obtained more voxels with higher classification accuracies when more changes in brain activation occurred across time points. In contrast, fewer changes or more similarities in brain activation across time points reduced classification accuracies. As a result, these findings suggested that patients with SCD who received hydroxyurea treatment had less severe changes in brain activation during working memory processing. This phenomenon could be an effect of hydroxyurea treatment maintaining working memory function (preventing its degradation) over time in patients with SCD. Therefore, changes in behavior performance in the 2-back task with hydroxyurea treatment were not detected at 1-year follow-up.

## 4.2 Increased brain activation during working memory processing

In addition to identifying more changes in brain activation during working memory processing in the non-HU control group at 1-year follow-up, we observed increased brain activation in 6 (the right crus I cerebellum, right inferior parietal lobe, right inferior temporal lobe, right angular gyrus, left cuneus and left middle frontal gyrus) of the 9 selected clusters in patients who did not receive hydroxyurea treatment but only in 1 cluster (the right superior parietal lobe) in the HU group at the same time point. Larger increases in BOLD signals were observed

in the non-HU control group as the working memory load increased from 0- to 1-back compared to from 1- to 2-back. These observations in the non-HU control group suggested an increase in cognitive effort during working memory processing in patients with SCD who did not receive hydroxyurea treatment at 1-year follow-up and as working memory load increased from 0- to 1-back. An elevation in brain activation in the salience network as a function of increasing demands of working memory tasks was reported by Engström et al. (2013). They observed a steeper increase of the BOLD signal as a function of increasing cognitive effort in participants with low working memory capacity. Although BOLD signals in the non-HU control group did not increase further as the working memory load increased from 1- to 2-back, they increased continuously in the HU group with increasing working memory load from 0- to 2-back. This coincides with the findings from the earlier study by Engström et al., (2013) in which they showed that high working memory performers have BOLD responses modulated by effort along a larger dynamic range.

In addition, some of the selected clusters contain key brain regions related to attention, working memory, and effortful processing, such as the inferior parietal lobe, right Crus I cerebellum, and right angular gyrus. For example, cluster ID 1 was located partially at the right Crus I of the cerebellum and right inferior occipital lobe. The Crus I of the cerebellum contributes to working memory [46, 47], and the occipital lobe has a role in object recognition and memory formation [48]. Cluster ID 3 was located in the right inferior parietal lobe and the right angular gyrus. Both brain regions are involved in spatial attention [49] and memory [50]. Cluster ID 1T was in the right superior frontal gyrus and middle frontal gyrus. The superior frontal gyrus is a key component in the neural network of working memory [51], and the middle frontal gyrus has a role in reorienting attention [52]. Therefore, in this study, we found increased BOLD signals at 1-year follow-up in brain regions associated with attention, working memory, and effortful processing that were potentially due to cognitive degradation with no hydroxyurea treatment in patients with SCD.

## 4.3 Limitations and future study

One limitation of this study was the small number of participants with complete fMRI and behavior data of n-back tasks at both time points. Due to the restricted size of the participant cohort, we assumed an exploratory framework for this study. To substantiate this study's outcomes, further validation in a larger cohort is needed. Moreover, the 2-back task is more difficult than the 1- and 0-back tasks; thus, some participants had problems understanding the task and failed to perform it per instructions. The participants' baseline working memory and some neurocognitive measures before treatment were slightly different and might have confounded the results. Baseline measurements were obtained at various ages because participants started hydroxyurea treatment at different ages; the severity of SCD also differed among the participants. Throughout the enrollment phase, the following indications of severity warranted the initiation of hydroxyurea treatment: the occurrence of 3 or more vaso-occlusive pain episodes necessitating emergency room visitation or hospitalization within the preceding year or a medical history of 2 or more episodes of acute chest syndrome. In light of these limitations, we conducted multivariate pattern analysis, which has higher sensitivity, even with smaller sample sizes. Traditional fMRI univariate analysis has a limited ability to identify subthreshold brain signals; thus, multivariate pattern analysis compensates for that limitation by analyzing brain signals in a multidimensional space, which has the potential for novel discoveries [53].

Hydroxyurea is a disease-modifying treatment, and patients need to take the drug throughout their lifetime for continued effect. Therefore, a longitudinal study that evaluates neurocognitive changes over a longer period is essential for obtaining more conclusive outcomes.

Furthermore, transformative treatments, like gene therapy [54] and hematopoietic cell transplantation [55], may produce more positive neurocognitive effects than hydroxyurea treatment. Therefore, future studies may investigate brain-activation patterns and neurocognitive changes during n-back tasks and other cognitive tasks in patients who receive transformative or other curative treatments.

## 5. Conclusion

Our present study suggested that patients who received hydroxyurea treatment for 1 year showed stability in working memory processing. It also identified brain regions with changes in activation between time points during the 2-back task via searchlight-pattern classification. Patients in the non-HU control group showed more brain regions with increased BOLD signals across time points and as working memory load increased from 0- to 1-back at 1-year follow-up. Elevated BOLD signals in these patients suggested an increase in cognitive effort during working memory processing. In contrast, patients in the HU group showed smaller change in brain activation during working memory processing across time points and fewer brain regions with increased BOLD signals at 1-year follow-up. As working memory increased, BOLD signals in the HU group increased continuously from 0- to 2-back but the increased BOLD signals in the non-HU control group were mostly observed from 0- to 1-back. Together, these results showed that hydroxyurea maintains work memory over time in patients with SCD.

## Supporting information

**S1 Table. Clinical hematologic values of the HU group at baseline and 1-year follow-up.** One-sample $t$-test or Wilcoxon signed rank test was used to compare 1-year follow-up to baseline; $p$ indicates significance (two-sided). Table was regenerated with permission from Wang et al. (2021).
(DOCX)

**S2 Table. Seven clusters (peak accuracy $\geq$.7 & volume $\geq$520 mm$^3$) were selected from the results of searchlight analysis of the non-HU control group for downstream analysis.**
(DOCX)

**S3 Table. Two clusters (peak accuracy $\geq$.7 & volume $\geq$520 mm$^3$) were selected from the results of searchlight analysis of the HU group for downstream analysis.**
(DOCX)

## Acknowledgments

We acknowledge Dr. Robert Ogg for providing assistance during the study and Dr. Angela J. McArthur for editing the manuscript.

## Author Contributions

**Conceptualization:** Jesyin Lai, Ping Zou, Josue L. Dalboni da Rocha, Andrew M. Heitzer, Matthew A. Scoggins, Akshay Sharma, Winfred C. Wang, Kathleen J. Helton, Ranganatha Sitaram.

**Data curation:** Ping Zou.

**Formal analysis:** Jesyin Lai, Ping Zou, Tushar Patni, Yimei Li.

**Funding acquisition:** Kathleen J. Helton.

**Investigation:** Jesyin Lai, Ping Zou, Josue L. Dalboni da Rocha.

**Methodology:** Jesyin Lai, Ping Zou, Josue L. Dalboni da Rocha, Andrew M. Heitzer, Matthew A. Scoggins, Akshay Sharma, Winfred C. Wang.

**Project administration:** Kathleen J. Helton, Ranganatha Sitaram.

**Supervision:** Ranganatha Sitaram.

**Visualization:** Jesyin Lai.

**Writing – original draft:** Jesyin Lai.

**Writing – review & editing:** Jesyin Lai, Ping Zou, Josue L. Dalboni da Rocha, Andrew M. Heitzer, Tushar Patni, Yimei Li, Matthew A. Scoggins, Akshay Sharma, Winfred C. Wang, Kathleen J. Helton, Ranganatha Sitaram.

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
