## [Decision Letter · Decision Letter 0]

14 Feb 2024

PONE-D-23-36219Hydroxyurea maintains working memory function in pediatric sickle cell diseasePLOS ONE

Dear Dr. Lai,

Thank you for submitting your manuscript to PLOS ONE. After careful consideration, we feel that it has merit but does not fully meet PLOS ONE’s publication criteria as it currently stands. Therefore, we invite you to submit a revised version of the manuscript that addresses the points raised during the review process.

**The reviewers found the study to be a very important area of investigation. However, there were also several areas that they identified for improvement, such as a better description of the control population and explanation for how and why certain patients were excluded from the analysis.**

We look forward to receiving your revised manuscript.

Kind regards,

Santosh L. Saraf

Academic Editor

PLOS ONE

This work was supported by the Cancer Center Support (CORE) grant CA21765 from the National Cancer Institute, grant RR029005 from the National Center for Research Resources, and ALSAC. Andrew M. Heitzer was supported by K23HL166697 (National Heart, Lung, and Blood Institute) during the time of this study.

I have read the journal's policy and the authors of this manuscript have the following competing interests: Dr. Akshay Sharma has received consultant fees from Spotlight Therapeutics, Medexus Inc., Vertex Pharmaceuticals, Sangamo Therapeutics, and Editas Medicine. He has also received research funding from CRISPR Therapeutics and honoraria from Vindico Medical Education. Dr. Sharma is the St. Jude Children’s Research Hospital site principal investigator of clinical trials for genome editing of sickle cell disease sponsored by Vertex Pharmaceuticals/CRISPR Therapeutics (NCT03745287), Novartis Pharmaceuticals (NCT04443907) and Beam Therapeutics (NCT05456880). The industry sponsors provide funding for the clinical trial, which includes salary support paid to Dr. Sharma’s institution. Dr. Sharma has no direct financial interest in these therapies. All other authors declare that the research was conducted in the absence of any commercial or financial relationships that could be construed as a potential conflict of interest. Dr. Andrew M. Heitzer has received consulting fees from Global Blood Therapeutics.

We note that one or more of the authors are employed by a commercial company.

“The funder provided support in the form of salaries for authors, but did not have any additional role in the study design, data collection and analysis, decision to publish, or preparation of the manuscript. The specific roles of these authors are articulated in the ‘author contributions’ section.”

Reviewers' comments:

Reviewer's Responses to Questions

**Comments to the Author**

1. Is the manuscript technically sound, and do the data support the conclusions?

Reviewer #1: Partly

Reviewer #2: Partly

2. Has the statistical analysis been performed appropriately and rigorously? 

Reviewer #1: I Don't Know

Reviewer #2: Yes

3. Have the authors made all data underlying the findings in their manuscript fully available?

Reviewer #1: Yes

Reviewer #2: Yes

4. Is the manuscript presented in an intelligible fashion and written in standard English?

Reviewer #1: Yes

Reviewer #2: Yes

5. Review Comments to the Author

Reviewer #1: In this manuscript, Lai and colleagues attempt to add to the growing body of literature of the neuroprotective effects of hydroxyurea, particularly when started in childhood. While I commend this effort as I think it is important to increase hydroxyurea use in individuals with SCD given its superiority as a disease-modifying therapy, I am concerned about the manuscript’s structure/content in its current form. Below are my major and minor concerns.

Major concerns

1. The abstract is far too long (the journal’s limit is 300 words) and currently reads like a results section. This should be a succinct summary of the manuscript that highlight the primary message of the study.

2. One of my primary concerns with this manuscript is that the authors do not appear to know their audience; although PLOS One is not specific to hematology, the majority of interested readers will be sickle cell providers/researchers. The parts of the introduction that focus on cognitive deficits in SCD and the current data regarding the neuroprotective effects of HU in SCD are incomplete/imprecise; I recommend having someone who is familiar with this literature review the introduction. As an example, the authors stated that “there is limited clinical neuroimaging data relevant to hydroxyurea” (page 4, lines 59-60). I recommend that the authors look at Fields et al., Blood 2019 for additional work in this area.

3. Similarly to above, the authors never actually explain an “n-back test”, which is arguably central to their study; although this is likely obvious to neuropsychologists, most hematologists are not aware of specific working memory tasks. The same holds true for the f-MRI pre-processing and searchlight analysis sections of the methods.

4. The last paragraph of the introduction is too long and is written like a methods section. This should be a summary of the introduction and lay out the primary hypothesis or objectives of the study.

5. The specific information on the control/study populations should be in the results section; however, more importantly is the mean age of these two groups. Cognitive deficts are well-known to have already started far before age 12 years. I also have to ask what percentage of the clinic’s population is actually on HU because that is a significant number of children who had not started the medication, despite the 2014 NHLBI Guidelines that recommend offering HU to all with the SCA genotypes as young as 9 months.

6. Going along with the above, although it is too late now, a more interesting study would have been to evaluate children with SCD who had been on HU for several years already compared to those not on HU. One of the flaws with the current design is that much of that year under study involved titrating the patient to MTD, i.e. the full HU effect could not be evaluated. Although the authors left out the lab values for the included patients, citing a paper by Wang et al., I would include that information in the results (a table comparing the controls versus HU group would be nice).

7. Unfortunately, it is difficult to interpret the results given the amount of excluded data in each group; data from up to 10 participants was excluded in the HU group and 3/11 in control group.

8. The results section 3.5 is long and confusing; I recommend a summary statement at the end of the paragraph explaining the main findings.

9. The first paragraph of the discussion is just re-hash of the results; this should instead be analysis of the findings and a review of the literature.

10. How the authors state their findings at present sound negative, ie that they did not observe an improvement in memory tasks across the year in those on HU; I would restate this as those who began HU had stability in their memory tasks whereas those in the control group had worsening.

Minor Concerns

1. The overall manuscript is lengthy and difficult to follow; I think editing (and therefore cutting down on its length) could make the message clearer.

2. Page 23, line 483- BMT and gene therapy are not curative; they are transformative therapies.

3. Tables 1&2 are needlessly complicated and do not add anything to the manuscript.

Reviewer #2: Lai et al hypothesized that hydroxyurea improves working memory function and impacts working memory processing in pediatric individuals with SCD. Additionally, specific regions of the brain with a known relation to working memory processing were assessed. Strengths include the longitudinal nature of this assessment and the methods of connecting tangible neurocognitive findings with imaging results. Overall, the article is well written and clearly describes the disease and neurocognitive issues encompassed there. The introduction additionally clarifies why the n-back testing is used/validated. It is easy for readers to follow with variable knowledge of neurocognitive deficits in SCD. The discussion clearly summarizes findings, relevance, and potential limitations. Recognizing the complexity of assessments performed in this manuscript, some effort should be put into clarifying results for ease of reader comprehension/to maintain consistency.

- Not all patients with SCD are included. It is worth noting why individuals with SC and SB+ were excluded. The rationale could be a lack of clear guidelines regarding HU use in these phenotypes. Authors can consider adding their rationale.

- According to Line 152 “Neurocognitive measures sensitive to deficits in SCD were used”. It would be nice to include a reference that validates this claim.

- Line 186 says 13 participants and 9 controls. What happened to the other 2 controls and the additional 7 on HU mentioned lines 103-108? A number of participants excluded for image quality is addressed in the paragraph after line 250. Please find a way to clarify a total number of participants and the number excluded earlier in the document for clarity. Perhaps including the number enrolled and the number excluded in the same place.

- In regards to the cut-off threshold for accuracy of 0.65 named in line 198, how was this cut-off point established? Is this validated in the literature for this testing? If so, please give a reference to clarify why 0.65 was used.

- Figure 3 Please address outliers or what was done in the analysis to account for these outliers. Consider adding characteristics of outliers or why there are so many in HU group. Address/rationale what is going on with why in 2-back reaction time went down for HU group.

- Figure 4 legend Line 289 define “MNI”

- Consider adding color key on figure 4 image (recognizing it is defined in the figure legend)

- Line 304 can you please explain the selection criteria of peak accuracy and volume that led the number in the analysis to be reduced from 27 to 11. Also, can you speak to the power of the analysis with over 50% of these clusters excluded? Or why excluded clusters were not relevant to the analysis. Additionally, accuracy elsewhere was measured at 0.65, and here it is 0.7

- Figure 6 legend (line 317) mentions excluding by clusters by “peak value” where previously it was stated clusters were excluded by “peak accuracy.” please clarify and keep language consistent for ease of understanding. Keep in mind changes in language may impact language used in table 1

- Figure 7B consider using a different legend specifically the 0-back in the HU group is difficult to distinguish from other time points

- Why is the N for figure 7A different for figure 7B for each group. What does this N represent in relation to the figure. That N is unclear as it was previously stated 7 clusters in non HU group and 2 in HU were examined. Is this N the number of patients who had this cluster distribution?

- Line 362 again, the n for each group has changed please explain.

- Can more information be added about the severity of SCD within each group? It was mentioned in the limitation section that the individuals had varying disease severity. It would help to add more information about this and how disease severity was determined.

- Also, can you add information regarding any significant CVA that may have occurred in the 1 year period in patients, if any? Any SCI in enrolled patients in 1 year study period?

- In methods clarify any procedure implemented to ensure HU was being used by study participants in the treatment group during that 1 year period.

- Adding a bolded declarative summary sentence at the beginning of each figure legend could help clarity throughout the manuscript

6. PLOS authors have the option to publish the peer review history of their article (what does this mean?). If published, this will include your full peer review and any attached files.

Reviewer #1: No

Reviewer #2: No

---

## [Author Response · Author response to Decision Letter 0]

1 May 2024

We thank the academic editor and reviewers for their thoughtful and constructive comments, which appear below in italics. We have addressed each comment, carefully revised the entire manuscript per the editor’s and reviewers’ recommendations and noted the location of those changes in the revised paper. 

Editor

Thank you for this information. We have reformatted our manuscript to meet PLOS ONE’s style requirements.

This work was supported by the Cancer Center Support (CORE) grant CA21765 from the National Cancer Institute, grant RR029005 from the National Center for Research Resources, and ALSAC. Andrew M. Heitzer was supported by K23HL166697 (National Heart, Lung, and Blood Institute) during the time of this study.

We have amended the Funding Statement accordingly by adding the recommended statement in the revised manuscript. We have also included the amended Funding Statement in the cover letter. 

I have read the journal's policy and the authors of this manuscript have the following competing interests: Dr. Akshay Sharma has received consultant fees from Spotlight Therapeutics, Medexus Inc., Vertex Pharmaceuticals, Sangamo Therapeutics, and Editas Medicine. He has also received research funding from CRISPR Therapeutics and honoraria from Vindico Medical Education. Dr. Sharma is the St. Jude Children’s Research Hospital site principal investigator of clinical trials for genome editing of sickle cell disease sponsored by Vertex Pharmaceuticals/CRISPR Therapeutics (NCT03745287), Novartis Pharmaceuticals (NCT04443907) and Beam Therapeutics (NCT05456880). The industry sponsors provide funding for the clinical trial, which includes salary support paid to Dr. Sharma’s institution. Dr. Sharma has no direct financial interest in these therapies. All other authors declare that the research was conducted in the absence of any commercial or financial relationships that could be construed as a potential conflict of interest. Dr. Andrew M. Heitzer has received consulting fees from Global Blood Therapeutics.

We note that one or more of the authors are employed by a commercial company.

“The funder provided support in the form of salaries for authors, but did not have any additional role in the study design, data collection and analysis, decision to publish, or preparation of the manuscript. The specific roles of these authors are articulated in the ‘author contributions’ section.”

We have included the recommended statement in the amended Funding Statement. We have also confirmed that the commercial affiliation does not alter our adherence to all PLOS ONE policies on sharing data and materials. We have included the amended Conflict of Interest Statement in the revised manuscript and the cover letter.

Reviewer #1

In this manuscript, Lai and colleagues attempt to add to the growing body of literature of the neuroprotective effects of hydroxyurea, particularly when started in childhood. While I commend this effort as I think it is important to increase hydroxyurea use in individuals with SCD given its superiority as a disease-modifying therapy, I am concerned about the manuscript’s structure/content in its current form. Below are my major and minor concerns.

Major concerns

1. The abstract is far too long (the journal’s limit is 300 words) and currently reads like a results section. This should be a succinct summary of the manuscript that highlight the primary message of the study.

Thank you for this comment. We have edited the Abstract to be more succinct and reduced the text to 296 words.

2. One of my primary concerns with this manuscript is that the authors do not appear to know their audience; although PLOS One is not specific to hematology, the majority of interested readers will be sickle cell providers/researchers. The parts of the introduction that focus on cognitive deficits in SCD and the current data regarding the neuroprotective effects of HU in SCD are incomplete/imprecise; I recommend having someone who is familiar with this literature review the introduction. As an example, the authors stated that “there is limited clinical neuroimaging data relevant to hydroxyurea” (page 4, lines 59-60). I recommend that the authors look at Fields et al., Blood 2019 for additional work in this area.

We have revised the Introduction and added the suggested work to that section of the paper (pages 5 & 6 of the tracked version). In addition, several co-authors of this study who are familiar with SCD and hydroxyurea have reviewed the literature cited in the Introduction.

3. Similarly to above, the authors never actually explain an “n-back test”, which is arguably central to their study; although this is likely obvious to neuropsychologists, most hematologists are not aware of specific working memory tasks. The same holds true for the f-MRI pre-processing and searchlight analysis sections of the methods.

We have added text explaining the n-back test in the Introduction (page 6 of the tracked version). For fMRI pre-processing, we used standard pre-processing steps in this study. We have added explanations and relevant references for these steps in the Methods (section 2.6). In addition, we have added sufficient details about searchlight analysis in the Methods (section 2.7). 

4. The last paragraph of the introduction is too long and is written like a methods section. This should be a summary of the introduction and lay out the primary hypothesis or objectives of the study.

We have removed the sentences related to Methods from the last paragraph of the Introduction (page 7 of the tracked version).

5. The specific information on the control/study populations should be in the results section; however, more importantly is the mean age of these two groups. Cognitive deficts are well-known to have already started far before age 12 years. I also have to ask what percentage of the clinic’s population is actually on HU because that is a significant number of children who had not started the medication, despite the 2014 NHLBI Guidelines that recommend offering HU to all with the SCA genotypes as young as 9 months.

We have moved the specific information about the study participants from the Methods to the Results (section 3.1). Concerning the comments on the age of participants starting hydroxyurea and the percentage of the clinic’s population on hydroxyurea therapy, we respectfully remind the reviewer that the protocol in question was completed before 2014 (it was started in 2011), and hydroxyurea was offered to most patients with SCD. Still, some were hesitant to take it and constituted the control population. Since the adoption of the NHLBI guidelines in 2014, and in line with our institutional efforts to increase hydroxyurea adherence, several programs have been established, and almost all eligible children are currently receiving either hydroxyurea or blood transfusions (as indicated).

6. Going along with the above, although it is too late now, a more interesting study would have been to evaluate children with SCD who had been on HU for several years already compared to those not on HU. One of the flaws with the current design is that much of that year under study involved titrating the patient to MTD (maximum tolerated dose), i.e. the full HU effect could not be evaluated. Although the authors left out the lab values for the included patients, citing a paper by Wang et al., I would include that information in the results (a table comparing the controls versus HU group would be nice).

Thank you for this comment. One limitation of the current report is that during the original study period, many children were not being treated on a steady-state maximum-tolerated dose of hydroxyurea. It is an excellent idea to compare neurocognitive treatment effects of children who have been on hydroxyurea for many years with those on other disease-modifying therapies. This is the subject of another ongoing study at St. Jude. The mean lab values for the HU group (e.g., Hgb, HbF, and absolute reticulocyte count) has been included in S1 Table in the Supporting Information section and can be referenced in the previous paper (Wang et al., 2021). A comparison of exit (1-year follow-up) and baseline values indicates an excellent response to hydroxyurea, despite the somewhat limited dosing period at the maximum-tolerated dose level. Unfortunately, we have difficulties retrieving the lab values for the non-HU group and are unable to include these for the non-HU group in the Supporting Information section. 

7. Unfortunately, it is difficult to interpret the results given the amount of excluded data in each group; data from up to 10 participants was excluded in the HU group and 3/11 in control group.

We apologize for this lack of information. Our original manuscript included typos regarding the number of participants in the HU group who were excluded. There were 4 participants excluded from the pre-treatment time point, and 3 participants excluded from the 1-year follow-up time point for the HU group. We have now corrected these typos. 

We had to exclude some participants’ data due to imaging-quality concerns. Because of the small number of participants, this study is exploratory in nature. We need to further validate the results of this study in a larger patient cohort. We have now clarified this in the first paragraph of the Limitations and future study (section 4.4) in the Discussion. Moreover, we moved sentences that describe the participant exclusions to the Methods (section 2.2), so that the total number of participants and the number of exclusions are in the same paragraph and presented earlier in the document. 

8. The results section 3.5 is long and confusing; I recommend a summary statement at the end of the paragraph explaining the main findings.

We have now edited the first paragraph of section 3.5 and added 2 sentences at the end to summarize the main findings.

9. The first paragraph of the discussion is just re-hash of the results; this should instead be analysis of the findings and a review of the literature.

We have edited the first paragraph of the Discussion to make it more succinct and added the relevant reference for the analysis of the findings.

10. How the authors state their findings at present sound negative, ie that they did not observe an improvement in memory tasks across the year in those on HU; I would restate this as those who began HU had stability in their memory tasks whereas those in the control group had worsening.

Thank you for this comment. We have edited the sentences and restated the findings, as suggested by the reviewer. The relevant changes can be found in the Abstract, the first paragraph of the Discussion (section 4.1), and the Conclusion. 

Minor Concerns

1. The overall manuscript is lengthy and difficult to follow; I think editing (and therefore cutting down on its length) could make the message clearer.

We have reviewed our manuscript and reduced its length. We also have had the paper reviewed by a scientific editor to improve the quality of the manuscript.

2. Page 23, line 483- BMT and gene therapy are not curative; they are transformative therapies.

We have now changed “curative” to “transformative,” as suggested by the reviewer.

3. Tables 1&2 are needlessly complicated and do not add anything to the manuscript.

We respectfully disagree with this comment from the reviewer. For the neuroimaging study, Tables 1 and 2 provide the details and locations of the identified clusters in the brain. However, we have moved these graphics to the Supporting Information section (S2 & S3 Tables) to reduce the length of the paper and so they can be reviewed by those interested in that information. 

Reviewer #2

Lai et al hypothesized that hydroxyurea improves working memory function and impacts working memory processing in pediatric individuals with SCD. Additionally, specific regions of the brain with a known relation to working memory processing were assessed. Strengths include the longitudinal nature of this assessment and the methods of connecting tangible neurocognitive findings with imaging results. Overall, the article is well written and clearly describes the disease and neurocognitive issues encompassed there. The introduction additionally clarifies why the n-back testing is used/validated. It is easy for readers to follow with variable knowledge of neurocognitive deficits in SCD. The discussion clearly summarizes findings, relevance, and potential limitations. Recognizing the complexity of assessments performed in this manuscript, some effort should be put into clarifying results for ease of reader comprehension/to maintain consistency.

- Not all patients with SCD are included. It is worth noting why individuals with SC and SB+ were excluded. The rationale could be a lack of clear guidelines regarding HU use in these phenotypes. Authors can consider adding their rationale.

Patients with SC and SB+ generally have milder disease and less severe symptoms. Specifically, their neurocognitive functions may differ from those with HbSS. Hence, to have a more homogenous population, patients with SC and SB+ disease were excluded from this study. We have added the reasons for not including SC and SB+ and the rationale for having SS and SB0 in our study in the Methods (section 2.1). 

- According to Line 152 “Neurocogni

---

## [Editor Report · Decision Letter 1]

9 May 2024

Hydroxyurea maintains working memory function in pediatric sickle cell disease

PONE-D-23-36219R1

Dear Dr. Lai,

We’re pleased to inform you that your manuscript has been judged scientifically suitable for publication and will be formally accepted for publication once it meets all outstanding technical requirements.

Kind regards,

Santosh L. Saraf

Academic Editor

PLOS ONE

---

## [Editor Report · Acceptance letter]

18 Jun 2024

PONE-D-23-36219R1 

PLOS ONE

Dear Dr. Lai, 

I'm pleased to inform you that your manuscript has been deemed suitable for publication in PLOS ONE. Congratulations! Your manuscript is now being handed over to our production team.

Kind regards, 

on behalf of

Dr. Santosh L. Saraf 

Academic Editor

PLOS ONE